# A New Coding Paradigm for the Primitive Relay Channel †

**Marco Mondelli** [1,*] **, S. Hamed Hassani** [2] **and Rüdiger Urbanke** [3]

1    Institute of Science and Technology (IST) Austria, 3400 Klosterneuburg, Austria
2    Department of Electrical and Systems Engineering, University of Pennsylvania,
     Philadelphia, PA 19104, USA; hassani@seas.upenn.edu
3    School of Computer and Communication Sciences, EPFL, CH-1015 Lausanne, Switzerland;
     ruediger.urbanke@epfl.ch
*    Correspondence: marco.mondelli89@gmail.com or marco.mondelli@ist.ac.at
†    This paper is an extended version of our paper published in 2018 IEEE International Symposium on
     Information Theory (ISIT 2018), Vail, CO, USA, 17–22 June 2018.

**Abstract:** We consider the primitive relay channel, where the source sends a message to the relay and to the destination, and the relay helps the communication by transmitting an additional message to the destination via a separate channel. Two well-known coding techniques have been introduced for this setting: decode-and-forward and compress-and-forward. In decode-and-forward, the relay completely decodes the message and sends some information to the destination; in compress-and-forward, the relay does not decode, and it sends a compressed version of the received signal to the destination using Wyner–Ziv coding. In this paper, we present a novel coding paradigm that provides an improved achievable rate for the primitive relay channel. The idea is to combine compress-and-forward and decode-and-forward via a chaining construction. We transmit over pairs of blocks: in the first block, we use compress-and-forward; and, in the second block, we use decode-and-forward. More specifically, in the first block, the relay does not decode, it compresses the received signal via Wyner–Ziv, and it sends only part of the compression to the destination. In the second block, the relay completely decodes the message, it sends some information to the destination, and it also sends the remaining part of the compression coming from the first block. By doing so, we are able to strictly outperform both compress-and-forward and decode-and-forward. Note that the proposed coding scheme can be implemented with polar codes. As such, it has the typical attractive properties of polar coding schemes, namely, quasi-linear encoding and decoding complexity, and error probability that decays at super-polynomial speed. As a running example, we take into account the special case of the erasure relay channel, and we provide a comparison between the rates achievable by our proposed scheme and the existing upper and lower bounds.

**Keywords:** primitive relay channel; compress-and-forward; decode-and-forward; chaining construction

## 1. Introduction

The relay channel, introduced by van der Meulen in [1], represents the simplest network model with a single source and a single destination. The source wants to communicate with the destination, and the relay helps the communication. More specifically, let $X_S$ be the signal sent by the source to the relay and to the destination, $Y_{SR}$ the signal received by the relay, $X_R$ the signal sent by the relay to the destination, and $Y_D$ the signal received by the destination which comes from the source and from the relay. Note that the relay channel has a broadcast component going from the source to the relay and to the destination, and a multiple access component going from the source and from the relay to the destination. The model is schematized in Figure 1.

Cover and El Gamal provided a general upper bound (the cut-set bound) and two lower bounds (decode-and-forward and compress-and-forward) in [2]. Since that seminal work, several lower bounds have been derived, i.e., amplify-and-forward, compute-and-forward, noisy network coding, quantize-map-and-forward, hybrid coding, see [3–7]. The cut-set bound is tight in most of the settings where capacity is known [2,8–10]. However, the cut-set bound was shown not be tight in some special cases [11,12], and novel upper bounds tighter than cut-set were recently presented in [13–16]. For a review on the relay channel, see also ([17] Chapter 16) and ([18] Chapter 9).

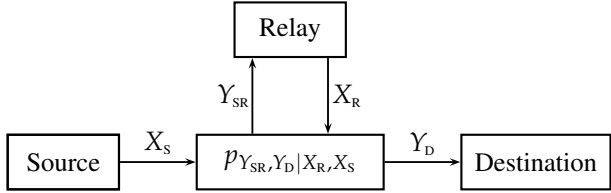

**Figure 1.** General relay channel.

Polar codes, introduced by Arıkan in [19], have been employed to devise practical schemes for the relay channel. In particular, for the case of the degraded relay channel where $X_S \to (X_R, Y_{SR}) \to Y_D$ forms a Markov chain, polar coding techniques for decode-and-forward are presented in [20–23]. Furthermore, for the case of the relay channel with orthogonal receiver components, a polar coding scheme for compress-and-forward is proposed in [22]. For general relay channels, polar coding techniques for decode-and-forward and compress-and-forward are described in [24]. We will adopt these schemes as primitives in our approach. Soft decode-and-forward relaying strategies which employ low-density parity-check (LDPC) codes are considered in [25].

In this work, we consider the relay channel with orthogonal receiver components, which is also known as the primitive relay channel. The difference with respect to the general relay channel consists in the fact that the destination receives two separate signals: $Y_{SD}$ from the source and $Y_{RD}$ from the relay. Basically, the multiple access component going from the source and from the relay to the destination is substituted by two parallel channels. Furthermore, we assume that the relay can listen and transmit simultaneously, namely, it is full-duplex. The model is schematized in Figure 2. Note that the relay communicates with the destination via a direct link. Thus, the relay can communicate reliably to the destination at a rate arbitrarily close to capacity by using a capacity achieving code (e.g., a random code or a polar code). Consequently, we can just assume that the relay and the destination are connected via a noiseless link of given capacity. Even in this simplified setting, the capacity of the primitive relay channel is unknown in general. A review on coding scheme for the primitive relay channel is contained in [26].

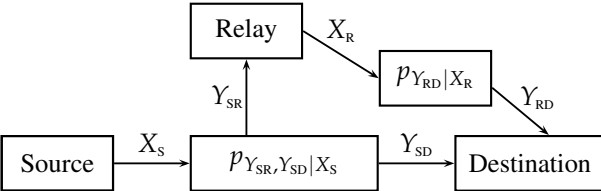

**Figure 2.** Primitive relay channel: relay channel with orthogonal receiver components.

The main contribution of this paper is a novel coding scheme that combines compress-and-forward with decode-and-forward and improves upon both of them. The idea is to consider pairs of blocks and use a chaining construction: in the first block, we perform a variation of compress-and-forward where the relay sends only a part of the compressed signal to the destination; in the second block, we perform decode-and-forward and the relay sends to the destination the new information bits together with the remaining part of the compressed signal coming from the previous block. The idea of chaining

was first presented in [27] to design universal codes and in [28] to guarantee strong security for the degraded wiretap channel. Since then, it has been employed in numerous other settings, such as the broadcast channel [29,30], the asymmetric channel [31,32], and the wiretap channel [33]. We highlight that our proposed coding paradigm is implementable with codes used for compress-and-forward and decode-and-forward. Thus, polar codes are an appealing choice [24]: they have an encoding and decoding complexity of $\Theta(n \log n)$ and a block error probability scaling roughly as $2^{-\sqrt{n}}$, where $n$ is the block length.

The rest of the paper is organized as follows. In Section 2, we provide a review of existing upper bounds (cut-set and its improvements) and lower bounds (direct transmission, decode-and-forward, partial decode-and-forward, compress-and-forward, and partial decode-compress-and-forward). These bounds are also evaluated for the special case of the erasure relay channel, which serves as a running example throughout the paper. In Section 3, we state and prove our new lower bound. In Section 4, we present some numerical results for the erasure relay channel: we compare the rates achieved by our proposed coding scheme with existing upper and lower bounds. Some concluding remarks are provided in Section 5. This work is an extended version of [34].

## 2. Existing Upper and Lower Bounds

We assume that all channels are binary memoryless and symmetric (BMS). We denote by $h_2(x) = -x \log_2 x - (1-x) \log_2(1-x)$ the binary entropy function and by $\mathcal{X}_S$, $\mathcal{X}_R$, $\mathcal{Y}_{SR}$, and $\mathcal{Y}_{SD}$ the alphabets associated with $X_S$, $X_R$, $Y_{SR}$, and $Y_{SD}$, respectively. We define $a \circ b = a + b(1-a)$ for any $a, b \in \mathbb{R}$.

Throughout the paper, we will use as a running example the special case of the erasure relay channel. As schematized in Figure 3, in the erasure relay channel, the links between source and destination and between source and relay are binary erasure channels (BECs) with erasure probabilities $\varepsilon_{SD}$ and $\varepsilon_{SR}$, respectively.

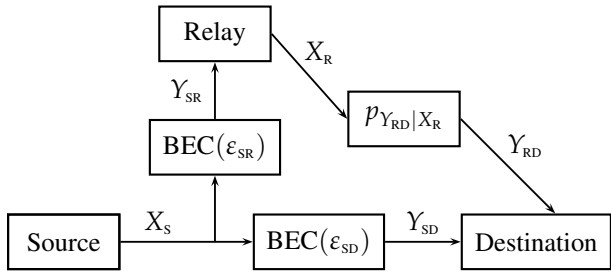

**Figure 3.** The erasure relay channel: primitive relay channel in which the link from source to relay is a BEC($\varepsilon_{SR}$) and the link from source to destination is a BEC($\varepsilon_{SD}$).

### 2.1. Cut-Set Upper Bound

For the *general relay channel*, the cut-set upper bound on the achievable rate $R$ is given by ([17] Theorem 16.1)

$$R \leq \max_{p_{X_S, X_R}} \min\{I(X_S, X_R; Y_D); I(X_S; Y_{SR}, Y_D | X_R)\}. \tag{1}$$

For the case of the *primitive relay channel*, the cut-set bound specializes to ([26] Proposition 1)

$$R \leq \max_{p_{X_S}} \min\{I(X_S; Y_{SD}) + C_{RD}; I(X_S; Y_{SR}, Y_{SD})\}. \tag{2}$$

For the special case of the *erasure relay channel*, the cut-set bound can be rewritten as

$$R \leq \min\{1 - \varepsilon_{SD} + C_{RD}; 1 - \varepsilon_{SR}\varepsilon_{SD}\}. \tag{3}$$

## 2.2. Improvements on a Cut-Set Upper Bound

For the case of the *primitive relay channel*, an upper bound demonstrating an explicit gap to the cut-set bound was presented in [13]. Furthermore, two new upper bounds that are generally tighter than cut-set are proposed in [14] for the symmetric primitive relay channel, in which $Y_{SR}$ and $Y_{SD}$ are conditionally identically distributed given $X_s$. The results of [14] are extended to the non-symmetric case and to the Gaussian case in [15,16], respectively.

Let us now state the result in ([15] Theorem 3.1), which provides an extension of the first bound of [14]. If a rate $R$ is achievable, then there exists some $p_{X_s}(x_s)$ and $a \geq 0$ such that

$$
\begin{cases}
R \leq I(X_s; Y_{SR}, Y_{SD}), \\[2ex]
R \leq I(X_s; Y_{SD}) + C_{RD} - a, \\[2ex]
R \leq I(X_s; Y_{SD}, \tilde{Y}_{SR}) + h_2\left(\sqrt{\dfrac{a \ln 2}{2}}\right) + \sqrt{\dfrac{a \ln 2}{2}} \log_2(|\mathcal{Y}_{SR}| - 1) - a,
\end{cases}
\tag{4}
$$

for any random variable $\tilde{Y}_{SR}$ with the same conditional distribution as $Y_{SR}$ given $X_s$. The evaluation of the term $I(X_s; Y_{SD}, \tilde{Y}_{SR})$ that gives the tightest bound is simple in the following special cases:

1. *Symmetric* ($Y_{SR}$ and $Y_{SD}$ are conditionally identically distributed given $X_s$): $I(X_s; Y_{SD}, \tilde{Y}_{SR}) = I(X_s; Y_{SD})$.
2. *Degraded* ($Y_{SD}$ is a stochastically degraded version of $Y_{SR}$): $I(X_s; Y_{SD}, \tilde{Y}_{SR}) = I(X_s; Y_{SR})$.
3. *Reversely degraded* ($Y_{SR}$ is a stochastically degraded version of $Y_{SD}$): $I(X_s; Y_{SD}, \tilde{Y}_{SR}) = I(X_s; Y_{SD})$.

For the special case of the erasure relay channel, the bound can be re-written as

$$
R \leq \max_{a \geq 0} \min\left\{1 - \varepsilon_{SR}\varepsilon_{SD}, 1 - \varepsilon_{SD} + C_{RD} - a, 1 - \min\{\varepsilon_{SR}, \varepsilon_{SD}\} + h_2\left(\sqrt{\frac{a \ln 2}{2}}\right) + \sqrt{\frac{a \ln 2}{2}} - a\right\}. \tag{5}
$$

In order to present the second bound of [14], we need some preliminary definitions. Given a channel transition probability $p(\omega|x)$, for any $p(x)$ and $d \geq 0$, we define $\Delta(p(x), d)$ as

$$
\Delta(p(x), d) = \max_{\tilde{p}(\omega|x)} \left( H(\tilde{p}(\omega|x)|p(x)) + D(\tilde{p}(\omega|x)||p(\omega|x)|p(x)) - H(p(\omega|x)|p(x)) \right), \tag{6}
$$

subject to the condition

$$
\frac{1}{2} \sum_{(x,\omega)} |p(x)\tilde{p}(\omega|x) - p(x)p(\omega|x)| \leq d, \tag{7}
$$

where $D(\tilde{p}(\omega|x)||p(\omega|x)|p(x))$ is the conditional relative entropy defined as

$$
D(\tilde{p}(\omega|x)||p(\omega|x)|p(x)) = \sum_{(x,\omega)} p(x)\tilde{p}(\omega|x) \log_2 \frac{\tilde{p}(\omega|x)}{p(\omega|x)}. \tag{8}
$$

$H(\tilde{p}(\omega|x)|p(x))$ is the conditional entropy defined with respect to the joint distribution $p(x)\tilde{p}(\omega|x)$, i.e.,

$$
H(\tilde{p}(\omega|x)|p(x)) = - \sum_{(x,\omega)} p(x)\tilde{p}(\omega|x) \log_2 \tilde{p}(\omega|x), \tag{9}
$$

and $H(p(\omega|x)|p(x))$ is the conditional entropy similarly defined with respect to $p(x)p(\omega|x)$. At this point, we can state the result in ([14] Theorem 4.2). If a rate $R$ is achievable, then there exists some $p_{X_S}(x_S)$ and $a \in [0, \min\{C_{RD}, H(Y_{SR} \mid X_S)\}]$ such that

$$
\begin{cases}
R \leq I(X_S; Y_{SR}, Y_{SD}), \\
R \leq I(X_S; Y_{SD}) + C_{RD} - a, \\
R \leq I(X_S; Y_{SD}) + \Delta\left(p_{X_S}(x_S), \sqrt{\dfrac{a \ln 2}{2}}\right).
\end{cases}
\tag{10}
$$

As pointed out at the end of Section IV.C of [14], for the special case of the symmetric erasure relay channel, we have that $\Delta(p_{X_S}(x_S), d) = \infty$ for all $p_{X_S}(x_S)$ and $d > 0$. Thus, formula (10) reduces to the cut-set bound (3).

### 2.3. Direct Transmission Lower Bound

In the direct transmission, the source communicates with the destination by using an optimal point-to-point code. The relay transmission is fixed at the most favorable symbol for the channel from the source to the destination.

For the *general relay channel*, direct transmission allows for achieving the following rate ([17] Section 16.3):

$$
R_{DT} = \max_{p_{X_S}, x_R} I(X_S; Y_D | X_R = x_R).
\tag{11}
$$

For the case of the *primitive relay channel*, the direct transmission lower bound specializes to

$$
R_{DT} = \max_{p_{X_S}} I(X_S; Y_{SD}).
\tag{12}
$$

Note that the direct transmission lower bound (12) meets the cut-set upper bound (2), and it equals the capacity of the primitive relay channel when either of the following two conditions holds:

1. the primitive relay channel is reversely degraded, which implies that $I(X_S; Y_{SD}) = I(X_S; Y_{SR}, Y_{SD})$;
2. $C_{RD} = 0$.

For the special case of the *erasure relay channel*, the direct transmission lower bound can be rewritten as

$$
R_{DT} = 1 - \varepsilon_{SD}.
\tag{13}
$$

The direct transmission lower bound (13) meets the cut-set upper bound (3), and it equals the capacity of the erasure relay channel when either $1 - \varepsilon_{SD} = 1 - \varepsilon_{SR}\varepsilon_{SD}$ or $C_{RD} = 0$.

### 2.4. Decode-and-Forward Lower Bound

In decode-and-forward, the relay completely decodes the received sequence and cooperates with the source to communicate the message to the destination.

For the *general relay channel*, decode-and-forward allows for achieving the following rate ([17] Theorem 16.2):

$$
R_{DF} = \max_{p_{X_S}, X_R} \min\{I(X_S, X_R; Y_D), I(X_S; Y_{SR} | X_R)\}.
\tag{14}
$$

For the case of the *primitive relay channel*, the decode-and-forward lower bound specializes to ([26] Proposition 2)

$$
R_{DF} = \max_{p_{X_S}} \min\{I(X_S; Y_{SD}) + C_{RD}; I(X_S; Y_{SR})\}.
\tag{15}
$$

Note that the decode-and-forward lower bound (15) meets the cut-set upper bound (2) and is equal to the capacity of the primitive relay channel when either of the following two conditions holds:

1.   the primitive relay channel is degraded, which implies that $I(X_s; Y_{SR}) = I(X_s; Y_{SR}, Y_{SD})$;
2.   $I(X_s; Y_{SR}) \geq I(X_s; Y_{SD}) + C_{RD}$.

For the special case of the *erasure relay channel*, the decode-and-forward lower bound can be rewritten as

$$R_{DF} = \min\{1 - \varepsilon_{SD} + C_{RD}; 1 - \varepsilon_{SR}\}. \tag{16}$$

The decode-and-forward lower bound (16) meets the cut-set upper bound (3), and it equals the capacity of the erasure relay channel when either $1 - \varepsilon_{SR} = 1 - \varepsilon_{SR}\varepsilon_{SD}$ or $1 - \varepsilon_{SD} + C_{RD} \leq 1 - \varepsilon_{SR}$.

## 2.5. Partial Decode-and-Forward Lower Bound

In partial decode-and-forward, the relay decodes and sends to the destination only part of the received sequence.

For the *general relay channel*, partial decode-and-forward allows for achieving the following rate ([17] Theorem 16.3):

$$R_{pDF} = \max_{p_{U,X_S,X_R}} \min\{I(X_s, X_R; Y_D), I(U; Y_{SR}|X_R) + I(X_s; Y_D|X_R, U)\}, \tag{17}$$

where the cardinality of the alphabet associated with $U$ can be bounded as $|\mathcal{U}| \leq |\mathcal{X}_s| \cdot |\mathcal{X}_R|$. Note that $U$ is an auxiliary random variable that represents the part of the message decoded by the relay. By taking $U = X_s$, we recover the decode-and-forward lower bound (14). Furthermore, by taking $U = \varnothing$, we recover the direct transmission lower bound (11).

Note that the partial decode-and-forward lower bound (17) meets the cut-set upper bound (1) when the relay channel has orthogonal sender components, namely, the broadcast channel from the source to the relay and the destination is decoupled into two parallel channels.

For the case of the *primitive relay channel*, the partial decode-and-forward lower bound specializes to ([26] Equation (5))

$$R_{pDF} = \max_{p_{U,X_S}} \min\{I(X_s; Y_{SD}) + C_{RD}, I(U; Y_{SR}) + I(X_s; Y_{SD}|U)\}, \tag{18}$$

with $|\mathcal{U}| \leq |\mathcal{X}_s|$.

For the special case of the *erasure relay channel*, we show that partial decode-and-forward does not provide any improvement upon both direct transmission and decode-and-forward. After some simple calculations, one obtains that

$$
\begin{aligned}
I(X_s; Y_{SD}) &= H(X_s) - H(X_s|Y_{SD}) = H(X_s)(1 - \varepsilon_{SD}), \\
I(U; Y_{SR}) &= H(U) - H(U|Y_{SR}) \\
&= H(U) - \varepsilon_{SR}H(U) - (1 - \varepsilon_{SR})H(U|X_s) \\
&= (1 - \varepsilon_{SR})(H(X_s) - H(X_s|U)), \\
I(X_s; Y_{SD}|U) &= H(X_s|U) - H(X_s|U, Y_{SD}) \\
&= H(X_s|U)(1 - \varepsilon_{SD}).
\end{aligned}
\tag{19}
$$

Hence, by setting $\alpha = H(X_s)$ and $\beta = H(X_s|U)$, we can re-write (18) as

$$
\begin{aligned}
R_{pDF} &= \max_{0 \leq \beta \leq \alpha \leq 1} \min\{\alpha(1 - \varepsilon_{SD}) + C_{RD}, \alpha(1 - \varepsilon_{SR}) + \beta(\varepsilon_{SR} - \varepsilon_{SD})\} \\
&= \max_{0 \leq \beta \leq 1} \min\{(1 - \varepsilon_{SD}) + C_{RD}, (1 - \varepsilon_{SR}) + \beta(\varepsilon_{SR} - \varepsilon_{SD})\}.
\end{aligned}
\tag{20}
$$

On the one hand, if $\varepsilon_{SR} \geq \varepsilon_{SD}$, then the maximum is achieved by taking $\beta = 1$, and $R_{pDF} = 1 - \varepsilon_{SD} = R_{DT}$. On the other hand, if $\varepsilon_{SR} \leq \varepsilon_{SD}$, then the maximum is achieved by taking $\beta = 0$, and $R_{pDF} =$

$\min\{(1 - \varepsilon_{\text{SD}}) + C_{\text{RD}}, 1 - \varepsilon_{\text{SR}}\} = R_{\text{DF}}$. Consequently, no improvement is possible over both direct transmission and decode-and-forward.

*2.6. Compress-and-Forward Lower Bound*

In compress-and-forward, the relay does not attempt to decode the received sequence, but it sends a (possibly compressed) description of it, denoted by $\hat{Y}_{\text{SR}}$, to the destination. Since this description is correlated with the sequence received by the destination from the source, Wyner–Ziv coding is used to reduce the rate needed to communicate it to the destination.

For the *general relay channel*, compress-and-forward allows for achieving the following rate ([17] Theorem 16.4):

$$R_{\text{CF}} = \max_{p_{X_{\text{S}}} p_{X_{\text{R}}} p_{\hat{Y}_{\text{SR}}|X_{\text{R}}, Y_{\text{SR}}}} \min\{I(X_{\text{S}}, X_{\text{R}}; Y_{\text{D}}) - I(Y_{\text{SR}}; \hat{Y}_{\text{SR}}|X_{\text{S}}, X_{\text{R}}, Y_{\text{D}}), I(X_{\text{S}}; \hat{Y}_{\text{SR}}, Y_{\text{D}}|X_{\text{R}})\}, \tag{21}$$

where the cardinality of the alphabet associated with $\hat{Y}_{\text{SR}}$ can be bounded as $|\hat{\mathcal{Y}}_{\text{SR}}| \leq |\mathcal{X}_{\text{R}}| \cdot |\mathcal{Y}_{\text{SR}}| + 1$. This expression can be equivalently rewritten as ([17] Remark 16.3)

$$R_{\text{CF}} = \max_{p_{X_{\text{S}}} p_{X_{\text{R}}} p_{\hat{Y}_{\text{SR}}|X_{\text{R}}, Y_{\text{SR}}}} \{I(X_{\text{S}}; \hat{Y}_{\text{SR}}, Y_{\text{D}}|X_{\text{R}}) : I(Y_{\text{SR}}; \hat{Y}_{\text{SR}}|X_{\text{R}}, Y_{\text{D}}) \leq I(X_{\text{R}}; Y_{\text{D}})\}. \tag{22}$$

The bound is in general not convex, therefore it can be improved via time sharing.

For the case of the *primitive relay channel*, the compress-and-forward lower bound specializes to ([26] Proposition 3)

$$R_{\text{CF}} = \max_{p_{X_{\text{S}}} p_{\hat{Y}_{\text{SR}}|Y_{\text{SR}}}} \{I(X_{\text{S}}; \hat{Y}_{\text{SR}}, Y_{\text{SD}}) : I(Y_{\text{SR}}; \hat{Y}_{\text{SR}}|Y_{\text{SD}}) \leq C_{\text{RD}}\}, \tag{23}$$

with $|\hat{\mathcal{Y}}_{\text{SR}}| \leq |\mathcal{Y}_{\text{SR}}| + 1$.

Note that the compress-and-forward lower bound (23) meets the cut-set upper bound (2), and it equals the capacity of the primitive relay channel when $H(Y_{\text{SR}}|Y_{\text{SD}}) \leq C_{\text{RD}}$. Indeed, in this case, we can pick $\hat{Y}_{\text{SR}} = Y_{\text{SR}}$, namely, the relay performs Slepian–Wolf source coding. Therefore, $R_{\text{CF}} = I(X_{\text{S}}; Y_{\text{SR}}, Y_{\text{SD}})$, which is one of the two terms in the cut-set bound.

On the contrary, if $H(Y_{\text{SR}}|Y_{\text{SD}}) > C_{\text{RD}}$, then we can degrade $Y_{\text{SR}}$ into $\hat{Y}_{\text{SR}}$, namely, the relay performs a step of lossy source coding. The relay transmits this lossy description to the destination that can decode it successfully since $\hat{Y}_{\text{SR}}$ requires less bits than $Y_{\text{SR}}$. However, after that the destination has recovered $\hat{Y}_{\text{SR}}$, there is a penalty loss: we can achieve rates up to $I(X_{\text{S}}; \hat{Y}_{\text{SR}}, Y_{\text{SD}})$, instead of up to $I(X_{\text{S}}; Y_{\text{SR}}, Y_{\text{SD}})$.

For the case of the *erasure relay channel*, we have that

$$H(Y_{\text{SR}}|Y_{\text{SD}}) = h_2(\varepsilon_{\text{SR}}) + \varepsilon_{\text{SD}}(1 - \varepsilon_{\text{SR}}). \tag{24}$$

Hence, if $C_{\text{RD}} \geq h_2(\varepsilon_{\text{SR}}) + \varepsilon_{\text{SD}}(1 - \varepsilon_{\text{SR}})$, then the compress-and-forward lower bound meets the cut-set upper bound, and it equals the capacity of the erasure relay channel.

On the contrary, if $C_{\text{RD}} < h_2(\varepsilon_{\text{SR}}) + \varepsilon_{\text{SD}}(1 - \varepsilon_{\text{SR}})$, it is not easy to find the best choice of $\hat{Y}_{\text{SR}}$ even for this simple scenario. Following [25], let us assume that $\hat{Y}_{\text{SR}}$ is the output of an erasure-erasure channel (EEC) with erasure probability $\hat{\varepsilon}_{\text{R}}$ and input $Y_{\text{SR}}$. This means that, if $Y_{\text{SR}} = ?$, then $\hat{Y}_{\text{SR}} = ?$ with probability 1; if $Y_{\text{SR}} \in \{0, 1\}$, then $\hat{Y}_{\text{SR}} = ?$ with probability $\hat{\varepsilon}_{\text{R}}$ and $\hat{Y}_{\text{SR}} = Y_{\text{SR}}$ with probability $1 - \hat{\varepsilon}_{\text{R}}$. Consequently,

$$\begin{aligned} I(X_{\text{S}}; \hat{Y}_{\text{SR}}, Y_{\text{SD}}) &= H(X_{\text{S}}) - H(X_{\text{S}}|\hat{Y}_{\text{SR}}, Y_{\text{SD}}) \\ &= H(X_{\text{S}})(1 - (\hat{\varepsilon}_{\text{R}} \circ \varepsilon_{\text{SR}}) \cdot \varepsilon_{\text{SD}}). \end{aligned} \tag{25}$$

Clearly, $I(X_S; \hat{Y}_{SR}, Y_{SD})$ is maximized by setting $p_{X_S}$ to the uniform distribution. Furthermore,

$$
\begin{aligned}
H(\hat{Y}_{SR}|Y_{SR}, Y_{SD}) &= H(\hat{Y}_{SR}|Y_{SR}) = (1 - \varepsilon_{SR})h_2(\hat{\varepsilon}_R), \\
H(\hat{Y}_{SR}|Y_{SD}) &= h_2(\varepsilon_{SR} \circ \hat{\varepsilon}_R) + \varepsilon_{SD}(1 - \varepsilon_{SR} \circ \hat{\varepsilon}_R).
\end{aligned}
\tag{26}
$$

As a result, the rate (23) can be rewritten as

$$
R_{CF} = \max_{0 \leq \hat{\varepsilon}_R \leq 1} \{1 - (\hat{\varepsilon}_R \circ \varepsilon_{SR}) \cdot \varepsilon_{SD} : h_2(\varepsilon_{SR} \circ \hat{\varepsilon}_R) + \varepsilon_{SD}(1 - \varepsilon_{SR} \circ \hat{\varepsilon}_R) - (1 - \varepsilon_{SR})h_2(\hat{\varepsilon}_R) \leq C_{RD}\}.
\tag{27}
$$

*2.7. Partial Decode-Compress-and-Forward Lower Bound*

In partial decode-compress-and-forward, the relay decodes and sends to the destination part of the source message, and it also sends to the destination a compressed description of the remaining signal by Wyner–Ziv coding.

For the *general relay channel*, partial decode-compress-and-forward allows for achieving the following rate ([2] Theorem 7):

$$
R_{pDCF} = \max \min\{I(X_S; \hat{Y}_{SR}, Y_D|U, X_R) + I(U; Y_{SR}|V, X_R), I(X_S, X_R; Y_D) - I(Y_{SR}; \hat{Y}_{SR}|U, X_S, X_R, Y_D)\},
\tag{28}
$$

where the maximization is taken over all the joint probability density functions of the form

$$
p_{U,V,X_S,X_R,Y_{SR},\hat{Y}_{SR},Y_D} = p_V p_{U|V} p_{X_S|U} p_{X_R|V} \cdot p_{Y_{SR},Y_D|X_S,X_R} p_{\hat{Y}_{SR}|X_R,Y_{SR},U}
\tag{29}
$$

such that

$$
I(X_R; Y_D|V) \geq I(\hat{Y}_{SR}; Y_{SR}|U, X_R, Y_D).
\tag{30}
$$

Partial decode–compress-and-forward is a generalization of partial decode-and-forward and compress-and-forward. Futhermore, it can strictly improve on both, e.g., for the state-dependent orthogonal relay channel with state information available at the destination [35].

Let us consider the case of the *primitive relay channel* and pick $V = \emptyset$. Then, the partial decode-compress-and-forward lower bound specializes to

$$
\begin{aligned}
R_{pDCF} = \max \min\{&I(X_S; \hat{Y}_{SR}, Y_{SD}|U) + I(U; Y_{SR}), \\
&I(X_S; Y_{SD}) + C_{RD} - I(Y_{SR}; \hat{Y}_{SR}|U, X_S)\},
\end{aligned}
\tag{31}
$$

such that

$$
C_{RD} \geq I(\hat{Y}_{SR}; Y_{SR}|Y_{SD}, U).
\tag{32}
$$

## 3. Main Result

We are now ready to state our new lower bound for the primitive relay channel.

**Theorem 1.** *Consider the transmission over a primitive relay channel, where the source sends $X_S$ to the relay and the destination, the relay receives $Y_{SR}$ from the source, the destination receives $Y_{SD}$ from the source, and relay and destination are connected via a noiseless link with capacity $C_{RD}$. Furthermore, denote by $\hat{Y}_{SR}$ the compressed description of $Y_{SR}$ transmitted by the relay, and define $I_{\max} = \max\{0, I(X_S; Y_{SR}) - I(X_S; Y_{SD})\}$. Then, the following rate is achievable:*

$$
R_{new} = \frac{(C_{RD} - I_{\max})I(X_S; \hat{Y}_{SR}, Y_{SD}) + \max\{I(X_S; Y_{SR}), I(X_S; Y_{SD})\}(I(Y_{SR}; \hat{Y}_{SR} \mid Y_{SD}) - C_{RD})}{I(Y_{SR}; \hat{Y}_{SR} \mid Y_{SD}) - I_{\max}},
\tag{33}
$$

*for any joint distribution $p_{X_s} p_{\hat{Y}_{SR}|Y_{SR}}$ such that*

$$I(X_s; Y_{SR}) < I(X_s; Y_{SD}) + C_{RD}, \tag{34}$$

$$I(Y_{SR}; \hat{Y}_{SR}|Y_{SD}) \geq C_{RD}, \tag{35}$$

*and where $|\hat{\mathcal{Y}}_{SR}| \leq |\mathcal{Y}_{SR}| + 1$. Furthermore, the rate (33) can be achieved by a polar coding scheme with encoding/decoding complexity $\Theta(n \log n)$ and error probability $O(2^{-n^\beta})$ for any $\beta \in (0, 1/2)$, where n is the block length.*

**Remark 1.** *If (34) does not hold, then decode-and-forward achieves the cut-set bound, and it is optimal. Furthermore, if (35) does not hold, then our scheme reduces to compress-and-forward, and the achievable rate is given by (23). As we will see in the proof, we have two slightly different schemes for the cases (i) $I(X_s; Y_{SR}) \geq I(X_s; Y_{SD})$ and (ii) $I(X_s; Y_{SR}) < I(X_s; Y_{SD})$. Thus, introducing the term $I_{\max}$ allows us to write the achievable rate in a more compact form.*

**Remark 2.** *The proposed scheme can be thought of as a particular form of time-sharing between decode-and-forward and compress-and-forward: in the first block, we are performing (a variant of) compress-and-forward, and, in the second block, we are performing decode-and-forward. However, we allow different time-sharing strategies across different channels: in the channel from relay to destination, part of the compressed message of the first block is sent together with the message of the second block. This is different from the 'classical' way of implementing time-sharing, which can be realized through the partial decode-compress-and-forward scheme, as described for example in [35]. In [35], in the same block, a part of the message is processed according to the decode-and-forward scheme, and the remaining part is processed according to the compress-and-forward scheme. Therefore, it is not clear that the rate achievable by our scheme can also be achieved by partial decode-compress-and-forward. In fact, in the special case considered in the numerical simulations of Section 4, our achievable rate strictly improves upon partial decode-compress-and-forward.*

**Remark 3.** *The proposed scheme is based on a chaining construction. Chaining can be thought of as a form of block Markov encoding, where the joint distribution is over blocks of symbols (instead of being over a single symbol). As described in detail in the proof, at the relay, we generate the first block according to a first codebook; we repeat part of the first block into the second block; and we generate the rest of the second block according to a second codebook. Thus, the repetition of part of the first block into the second block can be interpreted as a particular joint distribution over pairs of blocks.*

The special case of the erasure relay channel is handled by the corollary below.

**Corollary 1.** *Consider the transmission over the erasure relay channel, where $Y_{SD}$ is obtained from $X_s$ via a $BEC(\varepsilon_{SD})$, $Y_{SR}$ is obtained from $X_s$ via a $BEC(\varepsilon_{SR})$, $\hat{Y}_{SR}$ is obtained from $Y_{SR}$ via an $EEC(\hat{\varepsilon}_R)$, and the relay is connected to the destination via a noiseless link with capacity $C_{RD}$. Then, the rate*

$$R_{\text{new}} = \frac{(C_{RD} - \max\{0, \varepsilon_{SD} - \varepsilon_{SR}\})(1 - (\hat{\varepsilon}_R \circ \varepsilon_{SR}) \cdot \varepsilon_{SD})}{h_2(\varepsilon_{SR} \circ \hat{\varepsilon}_R) + \varepsilon_{SD}(1 - \varepsilon_{SR} \circ \hat{\varepsilon}_R) - (1 - \varepsilon_{SR})h_2(\hat{\varepsilon}_R) - \max\{0, \varepsilon_{SD} - \varepsilon_{SR}\}}$$
$$+ \frac{\max\{1 - \varepsilon_{SR}, 1 - \varepsilon_{SD}\}(h_2(\varepsilon_{SR} \circ \hat{\varepsilon}_R) + \varepsilon_{SD}(1 - \varepsilon_{SR} \circ \hat{\varepsilon}_R) - (1 - \varepsilon_{SR})h_2(\hat{\varepsilon}_R) - C_{RD})}{h_2(\varepsilon_{SR} \circ \hat{\varepsilon}_R) + \varepsilon_{SD}(1 - \varepsilon_{SR} \circ \hat{\varepsilon}_R) - (1 - \varepsilon_{SR})h_2(\hat{\varepsilon}_R) - \max\{0, \varepsilon_{SD} - \varepsilon_{SR}\}} \tag{36}$$

*is achievable for any $\hat{\varepsilon}_R \in [0, 1]$ such that*

$$1 - \varepsilon_{SR} < 1 - \varepsilon_{SD} + C_{RD}, \tag{37}$$

$$h_2(\varepsilon_{SR} \circ \hat{\varepsilon}_R) + \varepsilon_{SD}(1 - \varepsilon_{SR} \circ \hat{\varepsilon}_R) - (1 - \varepsilon_{SR})h_2(\hat{\varepsilon}_R) \geq C_{RD}. \tag{38}$$

*Furthermore, the rate (36) can be achieved by a polar coding scheme with encoding/decoding complexity* $\Theta(n \log n)$ *and error probability* $O(2^{-n^\beta})$ *for any* $\beta \in (0, 1/2)$, *where n is the block length.*

The proof of Corollary 1 easily follows from the application of Theorem 1 and of Formulas (25) and (26). We will now proceed with the proof of our main result.

**Proof of Theorem 1.** We start by presenting the main idea of our scheme. We split the transmission into two blocks. In the first block, we perform a variant of compress-and-forward: the relay does not decode the received sequence, but it sends a compressed description of it to the destination. However, differently from standard compress-and-forward, we require that (35) holds. Hence, we cannot transmit all the compressed description $\hat{Y}_{SR}$ during the first block. In the second block, we perform decode-and-forward: the relay completely decodes the received sequence. Furthermore, we choose the length of the second block so that the relay can transmit the part of $\hat{Y}_{SR}$ that was not sent in the previous block plus the new information needed to decode the second block.

Let us now describe this scheme more in detail and provide the achievability proof of the rate (33). First, we deal with the case $I(X_S; Y_{SR}) \geq I(X_S; Y_{SD})$.

Consider the transmission of the first block. Denote by $n_1$ and $R_1$ the block length and the rate of the message transmitted by the source, and let $R_1$ approach from below $I(X_S; \hat{Y}_{SR}, Y_{SD})$. The relay receives $Y_{SR}$ and constructs the compressed description $\hat{Y}_{SR}$. Recall that the destination receives the side information $Y_{SD}$ from the source. Hence, by using Wyner–Ziv coding, the destination needs from the relay a number of bits approaching from above $I(Y_{SR}; \hat{Y}_{SR} \mid Y_{SD}) \cdot n_1$, in order to decode the message sent by the source. As $I(Y_{SR}; \hat{Y}_{SR} | Y_{SD}) \geq C_{RD}$, the relay transmits right away a number of these bits approaching from below $C_{RD} \cdot n_1$. The number of remaining bits approaches from above $(I(Y_{SR}; \hat{Y}_{SR} | Y_{SD}) - C_{RD}) \cdot n_1$, and it is stored by the relay. The destination stores the message received from the relay and the observation $Y_{SD}$ obtained from the source.

Consider the transmission of the second block and define

$$\alpha = \frac{I(Y_{SR}; \hat{Y}_{SR} | Y_{SD}) - C_{RD}}{C_{RD} - \left(I(X_S; Y_{SR}) - I(X_S; Y_{SD})\right)}. \tag{39}$$

Denote by $n_2$ and $R_2$ the block length and the rate of the message transmitted by the source. Let $n_2 = n_1 \cdot \alpha$ and let $R_2$ approach from below $I(X_S; Y_{SR})$. The relay receives $Y_{SR}$ and successfully decodes the message. Again, the destination receives the side information $Y_{SD}$ from the source. Hence, it needs from the relay a number of bits approaching from above $(I(X_S; Y_{SR}) - I(X_S; Y_{SD})) \cdot n_1 \cdot \alpha$, in order to decode the message sent by the source. The relay transmits to the destination these $(I(X_S; Y_{SR}) - I(X_S; Y_{SD})) \cdot n_1 \cdot \alpha$ information bits plus the $(I(Y_{SR}; \hat{Y}_{SR} | Y_{SD}) - C_{RD}) \cdot n_1$ bits remaining from the previous block. This transmission is reliable as (39) implies that

$$\left(I(X_S; Y_{SR}) - I(X_S; Y_{SD})\right) \cdot n_1 \cdot \alpha + \left(I(Y_{SR}; \hat{Y}_{SR} | Y_{SD}) - C_{RD}\right) \cdot n_1 = C_{RD} \cdot n_2. \tag{40}$$

At this point, the destination can reconstruct the second block by using the side information received from the source and the extra $(I(X_S; Y_{SR}) - I(X_S; Y_{SD})) \cdot n_1 \cdot \alpha$ bits received from the relay. Furthermore, it can also reconstruct the first block by using the side information previously received from the source and the extra $I(Y_{SR}; \hat{Y}_{SR} \mid Y_{SD}) \cdot n_1$ bits received from the relay (partly in the first and partly in the second block).

The overall block length is $n = n_1 + n_2 = (1 + \alpha)n_1$, and the achievable rate is

$$R = \frac{R_1 + \alpha R_2}{1 + \alpha}, \tag{41}$$

which approaches from below

$$\frac{\left(C_{\mathrm{RD}} - \left(I(X_{\mathrm{S}}; Y_{\mathrm{SR}}) - I(X_{\mathrm{S}}; Y_{\mathrm{SD}})\right)\right) I(X_{\mathrm{S}}; \hat{Y}_{\mathrm{SR}}, Y_{\mathrm{SD}})}{I(Y_{\mathrm{SR}}; \hat{Y}_{\mathrm{SR}} | Y_{\mathrm{SD}}) - \left(I(X_{\mathrm{S}}; Y_{\mathrm{SR}}) - I(X_{\mathrm{S}}; Y_{\mathrm{SD}})\right)} + \frac{\left(I(Y_{\mathrm{SR}}; \hat{Y}_{\mathrm{SR}} | Y_{\mathrm{SD}}) - C_{\mathrm{RD}}\right) I(X_{\mathrm{S}}; Y_{\mathrm{SR}})}{I(Y_{\mathrm{SR}}; \hat{Y}_{\mathrm{SR}} | Y_{\mathrm{SD}}) - \left(I(X_{\mathrm{S}}; Y_{\mathrm{SR}}) - I(X_{\mathrm{S}}; Y_{\mathrm{SD}})\right)}. \tag{42}$$

Note that the expression (42) coincides with (33) when $I(X_{\mathrm{S}}; Y_{\mathrm{SR}}) \geq I(X_{\mathrm{S}}; Y_{\mathrm{SD}})$.

The case $I(X_{\mathrm{S}}; Y_{\mathrm{SR}}) < I(X_{\mathrm{S}}; Y_{\mathrm{SD}})$ is handled in a similar way. As concerns the transmission of the first block, nothing changes. Denote by $n'_1$ and $R'_1$ the block length and the rate of the message transmitted by the source, and let $R'_1$ approach from below $I(X_{\mathrm{S}}; \hat{Y}_{\mathrm{SR}}, Y_{\mathrm{SD}})$. The relay receives $Y_{\mathrm{SR}}$ and constructs the compressed description $\hat{Y}_{\mathrm{SR}}$. By using Wyner–Ziv coding, the destination needs from the relay a number of bits approaching from above $I(Y_{\mathrm{SR}}; \hat{Y}_{\mathrm{SR}} \mid Y_{\mathrm{SD}}) \cdot n'_1$, in order to decode the message sent by the source. As $I(Y_{\mathrm{SR}}; \hat{Y}_{\mathrm{SR}} | Y_{\mathrm{SD}}) \geq C_{\mathrm{RD}}$, the relay transmits right away a number of these bits approaching from below $C_{\mathrm{RD}} \cdot n'_1$. The number of remaining bits approaches from above $\left(I(Y_{\mathrm{SR}}; \hat{Y}_{\mathrm{SR}} | Y_{\mathrm{SD}}) - C_{\mathrm{RD}}\right) \cdot n'_1$ and it is stored by the relay. The destination stores the message received from the relay and the observation $Y_{\mathrm{SD}}$ obtained from the source.

As concerns the transmission of the second block, define

$$\alpha' = \frac{I(Y_{\mathrm{SR}}; \hat{Y}_{\mathrm{SR}} | Y_{\mathrm{SD}}) - C_{\mathrm{RD}}}{C_{\mathrm{RD}}}, \tag{43}$$

and denote by $n'_2$ and $R'_2$ the block length and the rate of the message transmitted by the source. Let $n'_2 = n'_1 \cdot \alpha'$ and let $R'_2$ approach from below $I(X_{\mathrm{S}}; Y_{\mathrm{SD}})$. The relay discards the received message and transmits to the destination the $\left(I(Y_{\mathrm{SR}}; \hat{Y}_{\mathrm{SR}} | Y_{\mathrm{SD}}) - C_{\mathrm{RD}}\right) \cdot n'_1$ bits remaining from the previous block. This transmission is reliable as (43) implies that

$$\left(I(Y_{\mathrm{SR}}; \hat{Y}_{\mathrm{SR}} | Y_{\mathrm{SD}}) - C_{\mathrm{RD}}\right) \cdot n'_1 = C_{\mathrm{RD}} \cdot n'_2. \tag{44}$$

At this point, the destination can reconstruct the second block by using the message received from the source. Furthermore, it can also reconstruct the first block by using the side information previously received from the source and the extra $I(Y_{\mathrm{SR}}; \hat{Y}_{\mathrm{SR}} \mid Y_{\mathrm{SD}}) \cdot n_1$ bits received from the relay (partly in the first and partly in the second block).

The overall block length is $n' = n'_1 + n'_2 = (1 + \alpha')n'_1$ and the achievable rate is

$$R' = \frac{R'_1 + \alpha' R'_2}{1 + \alpha'}, \tag{45}$$

which approaches from below

$$\frac{C_{\mathrm{RD}} \cdot I(X_{\mathrm{S}}; \hat{Y}_{\mathrm{SR}}, Y_{\mathrm{SD}}) + \left(I(Y_{\mathrm{SR}}; \hat{Y}_{\mathrm{SR}} | Y_{\mathrm{SD}}) - C_{\mathrm{RD}}\right) I(X_{\mathrm{S}}; Y_{\mathrm{SD}})}{I(Y_{\mathrm{SR}}; \hat{Y}_{\mathrm{SR}} | Y_{\mathrm{SD}})}. \tag{46}$$

Note that the expression (46) coincides with (33) when $I(X_{\mathrm{S}}; Y_{\mathrm{SR}}) < I(X_{\mathrm{S}}; Y_{\mathrm{SD}})$.

Clearly, the coding scheme described so far can be implemented with codes that are suitable for compress-and-forward and for decode-and-forward. Hence, we can employ the polar coding schemes for compress-and-forward and for decode-and-forward presented in [24]. However, polar codes require block lengths $n_1$ and $n_2$ (or $n'_1$ and $n'_2$) that are powers of two, which puts a constraint on the possible values for $\alpha = n_2/n_1$ (or $\alpha' = n'_2/n'_1$). To remove this constraint and achieve the rate (33) for any $\alpha$, it suffices to use the punctured polar codes described in ([36] Theorem 1). □

## 4. Numerical Results

Let us consider the special case of the erasure relay channel. In Figure 4, we compare the achievable rate (36) of our scheme with the existing upper and lower bounds, i.e., the cut-set upper

bound (3) (which coincides with the improved bound (5)), the decode-and-forward lower bound (16) and the compress-and-forward lower bound (27). We consider two pairs of choices for $\varepsilon_{\text{SD}}$ and $\varepsilon_{\text{SR}}$: $(\varepsilon_{\text{SD}}, \varepsilon_{\text{SR}}) = (0.85, 0.5)$ for the plot on the left (see Figure 4), and $(\varepsilon_{\text{SD}}, \varepsilon_{\text{SR}}) = (0.4, 0.2)$ for the plot on the right. We plot the various bounds as functions of $C_{\text{RD}}$. Our scheme outperforms both decode-and-forward and compress-and-forward for an interval of values of $C_{\text{RD}}$ in both settings. As $C_{\text{RD}}$ increases, the improvement guaranteed by our strategy decreases, until eventually the performance of our scheme is matched by compress-and-forward.

For a general primitive relay channel, it is not immediate how to compare the partial decode-compress-and-forward rate given in (28) with our new rate given in (33)—the partial decode-compress-and-forward scheme involves three auxiliary random variables $(U, V, \hat{Y}_{\text{SR}})$ and the complex joint distribution expressed in (29) to maximize over. Thus, one immediate advantage of our new rate is that it is easier to compute. In fact, the proposed lower bound involves only one auxiliary random variable ($\hat{Y}_{\text{SR}}$). Even if we simplify the partial decode-compress-and-forward rate as in (31), the formula remains harder to evaluate (two auxiliary random variables: $U, \hat{Y}_{\text{SR}}$). Although a full optimization over all parameters is very challenging, we have specialized (31) to the setting of the erasure relay channel and, for fairness of comparison with the other schemes, we have considered the case in which $\hat{Y}_{\text{SR}}$ is obtained from $Y_{\text{SR}}$ via an EEC($\hat{\varepsilon}_{\text{R}}$). Then, by performing the maximization numerically over $\hat{\varepsilon}_{\text{R}}$ and over all the auxiliary random variables $U$ s.t. $|U| \leq 2$, the achievable rate of partial decode-compress-and-forward does not improve upon decode-and-forward and compress-and-forward. Therefore, in this setting, partial decode-compress-and-forward is strictly worse than our proposed scheme.

In [25], for $\varepsilon_{\text{SD}} = 0.85$, $\varepsilon_{\text{SR}} = 0.5$ and $C_{\text{RD}} = 0.99125$, the proposed soft decode-and-forward strategy based on LDPC codes achieves a rate of 0.507, while both decode-and-forward and compress-and-forward achieve a rate of 0.5. Our new coding strategy is reliable for rates up to 0.545, hence it outperforms all existing lower bounds. As a reference, note that in this setting the cut-set bound is 0.575.

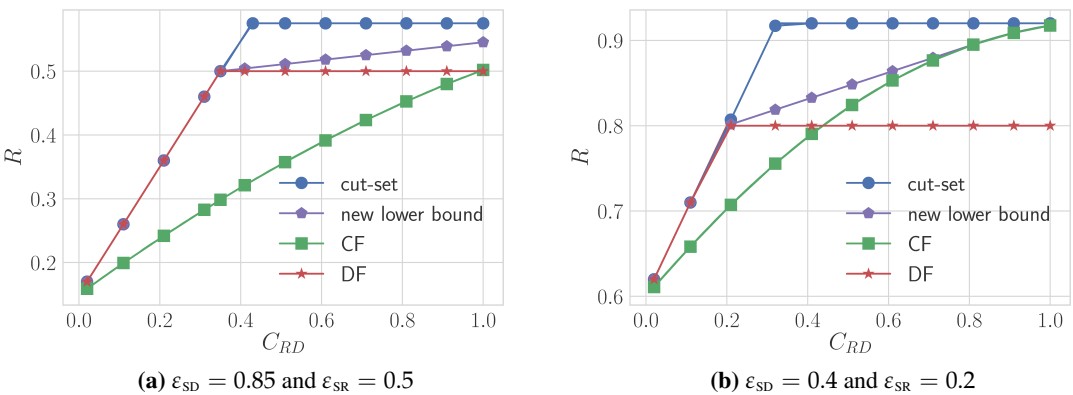

**(a)** $\varepsilon_{\text{SD}} = 0.85$ and $\varepsilon_{\text{SR}} = 0.5$　　　　　　　　　**(b)** $\varepsilon_{\text{SD}} = 0.4$ and $\varepsilon_{\text{SR}} = 0.2$

**Figure 4.** Comparison between the achievable rate provided by our strategy and the existing upper and lower bounds. We use "CF" and "DF" as abbreviations for "compress-and-forward" and "decode-and-forward", respectively.

## 5. Conclusions

We have proposed a new coding paradigm for the primitive relay channel that combines compress-and-forward and decode-and-forward by means of a chaining construction. The achievable rates obtained by our scheme surpass the state-of-the-art coding approaches (compress-and-forward, decode-and-forward, and the soft decode-and-forward strategy of [25]). Our coding paradigm is general in the sense that we treat decode-and-forward and compress-and-forward as existing primitives. For this reason, any coding scheme that can be used to implement decode-and-forward/compress-and-forward can also be used to implement our new strategy. Polar codes are one notable

example, since polar coding schemes for decode-and-forward and compress-and-forward have been developed; see [20–24]. This leads to a scheme with the typical attractive features of polar codes, i.e., quasi-linear encoding/decoding complexity and fast decay of the error probability. A detailed analysis of the finite length performance of polar codes for our strategy (as well as of polar codes for decode-and-forward and compress-and-forward) is an interesting direction for future research.

In the numerical simulations, we consider the special case of the erasure relay channel. In this setting, the upper bounds presented in Section 2.2 do not provide an improvement over the cut-set bound. An interesting avenue for future work is to study the performance of our strategy in scenarios where the cut-set bound is not tight (e.g., as in [11,12,35]). For example, in [35], the model also includes a state sequence, and the partial decode-compress-and-forward strategy crucially takes advantage of it by optimally adapting its transmission to the dependence of the orthogonal channels on the state sequence. In this paper, we do not consider such a state sequence, and it is not obvious how to adapt our results to the model of [35].

**Author Contributions:** Conceptualization, M.M., S.H.H. and R.U.; Formal analysis, M.M., S.H.H. and R.U.; Funding acquisition, M.M. and R.U.; Supervision, R.U.; Visualization, S.H.H.; Writing—review & editing, M.M.

**Funding:** M.M. was supported by an Early Postdoc.Mobility fellowship from the Swiss NSF. R.U. was supported by Grant No. 200021_156672/1 of the Swiss NSF.

**Conflicts of Interest:** The authors declare no conflict of interest.

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
