# Peer review of "A New Coding Paradigm for the Primitive Relay Channelâ€"

_algorithms, doi:10.3390/a12100218_

Round 1
Reviewer 1 Report
This is an interesting paper on a well-studied topic. Any contribution on the relay channel would have received significantly more interest about 10 years ago, but the problem studied here is fundamental, and can remain to be relevant for a long time.
Although the contribution of the paper is rather limited, it deserves to be published, if correct. The bulk of the paper is a review of existing upper and lower bounds on the capacity of a particular class of relay channels, called the primitive relay channel. The main contribution is a new achievable scheme. The benefit of this scheme is illustrated through an example of an erasure relay channel.
I have a few major issues that I believe must be addressed before this paper can be published:
- I am not fully convinced that the proposed scheme provides any improvement over the existing partial decode-compress-and-forward (pDCF) scheme. There is no argument on this in the paper, and for reasons not clear to me, the performance of the pDCF scheme on the example used for comparison is not included.
For the setting illustrated in Fig. 4(b), I believe the rate achieved by the proposed scheme can be achieved through time-sharing between DF and CF schemes for two different C_{RD} values, and hence, should be achievable by pDCF. (I wonder if similar will happen for Fig. 4(a) as well if C_{RD} is increased further ??).
- Interestingly, the authors refer to a paper [34], which exactly shows what they are claiming for their own scheme: that pDCF improves the rate compared to either DF or CF alone. Since the model in [34] is yet another primitive relay channel it only makes sense that the authors provide a comparison with pDCF, and possibly also in the setting studied in [34] (especially if that allows easy computation of the rate achieved by the proposed scheme).
- I don't particularly understand the merit of comparison with the code design in [25]. Does that code provide any improvement beyond pDCF? Since the proposed scheme is based on random coding arguments, I believe it makes sense to compare with other such results, and the claim that the proposed random coding rate can be approximated in practice (in some special cases) with polar codes should be sufficient.
- How is the idea of "chaining" different from block Markov encoding?
- One page is spent on explaining the upper bound in [14], that is tighter than cut-set, but in the end, for the case that is considered here, this bounds ends up reducing to cut-set. What is the point of all those derivations then? Since this bound is not derived here, and does not lead to any conclusions, I suggest they are removed.
The authors could instead try to identify other more general scenarios (than erasure relay channel) where this bound is tighter than cut-set, and maybe provides some tight (or, at least, tighter) bound. This seems to be the case in [11, 12, 34], which results in a much more interesting result.
Reviewer 2 Report
In this paper, the authors consider the classic primitive relay channel model and propose a novel coding scheme that combines compress-and-forward with decode-and-forward in order to improve upon both of them.
The main result is Theorem 1, in which the transmission rate achievable with the proposed coding scheme is derived, which results to be larger than that achievable by compress-and-forward and decode-and-forward used alone.
Despite the proposed coding scheme is somehow incremental, such a result looks novel to me. Therefore, despite the novelty of this paper is somehow limited, I believe that it may deserve publication.
I did not check the theoretical derivations carefully, but they look correct to me.
What indeed is a bit vague is the reference to polar codes as a possible solution for implementing the proposed scheme. They are mentioned several times, but only some asymptotic complexity expressions are provided. I believe that such a case study should be treated in mode detail and with more quantitative arguments, possibly considering some real cases in the finite length regime as well.
Another limitation of this study is that only the erasure channel model is considered, while the generality of the main result should be possibly demonstrated by considering a wider set of channel models.
Reviewer 3 Report
This paper improves upon the achievable rate in the classic (primitive) relay channel, by using a combination of two previously suggested results across two adjacent blocks; the compress-and-forward and the decode-and-forward one. The paper does a thorough job in surveying past results and explaining the problem. I have only a few minor comments and requests for clarifications, after which the paper can be accepted.
1. Introduction - It isn't clear why the assumption that the channel between the relay and the destination is noiseless does not limit the generality.
2. L. 102 - The expression 1-\eps_{SD}=1-\eps_{SR}\eps_{SD} seems to imply that \eps_{SR}=1 (unless \eps_{SD}=0). If that is so, isn't this case equivalent to having C_{RD}=0?
3. Eq. (33) - The expression max{0,I(X_S;Y_{SR})-I(X_S;Y_{SD})) appears three times. I would suggest to denote it by an extra variable, and perhaps provide some intuition about its role in the capacity of your scheme (alongside (34)).
Round 2
Reviewer 1 Report
I think the authors addressed my main concerns. I thank them for their detailed responses. I only ask the authors to revise the following statement:
"Therefore, in this setting, partial decode-compress-and-forward is strictly worse than our proposed scheme."
This is not accurate as the authors consider a simplified version of the partial decode-compress-and-forward scheme. This should be clarified.
I also suggest not to use ? for the erasure symbol above (25), but this is up to the authors.
Reviewer 2 Report
I am satisfied with how the authors addressed my concerns, and I believe that the revised paper can be accepted for publication.